# Recent Progress in Enhanced Cancer Diagnosis, Prognosis, and Monitoring Using a Combined Analysis of the Number of Circulating Tumor Cells (CTCs) and Other Clinical Parameters

**DOI:** 10.3390/cancers15225372

**Published:** 2023-11-11

**Authors:** Thi Ngoc Anh Nguyen, Po-Shuan Huang, Po-Yu Chu, Chia-Hsun Hsieh, Min-Hsien Wu

**Affiliations:** 1Graduate Institute of Biomedical Engineering, Chang Gung University, Taoyuan City 33302, Taiwan; d1131003@cgu.edu.tw (T.N.A.N.); leo_6813@cgu.edu.tw (P.-S.H.); d000018394@cgu.edu.tw (P.-Y.C.); 2Division of Hematology-Oncology, Department of Internal Medicine, New Taipei City Municipal TuCheng Hospital, New Taipei City 23652, Taiwan; wisdom5000@cgmh.org.tw; 3Division of Hematology-Oncology, Department of Internal Medicine, Chang Gung Memorial Hospital at Linkou, Taoyuan City 33302, Taiwan

**Keywords:** circulating tumor cells, CTCs, CTC counts, multiple parameter, clinical application, cancer diagnosis, cancer prognosis

## Abstract

**Simple Summary:**

Counting the number of circulating tumor cells (CTCs) in blood samples has been recognized as being clinically useful. However, challenges like the rarity and heterogeneity of CTCs have limited their widespread use in clinical practice. To address these challenges, a feasible direction is to combine the CTC counts with the routinely used clinical data relevant to cancer detection or screening for analysis. Recent studies demonstrate that this innovative approach has successfully improved cancer detection, prognosis, assessment, and the ability to differentiate between cancers at various stages and with different characteristics. The combination of CTC counts with clinical parameters represents a promising avenue for enhancing the clinical applicability of CTC analysis in cancer management.

**Abstract:**

Analysis of circulating tumor cells (CTCs) holds promise to diagnose cancer or monitor its development. Among the methods, counting CTC numbers in blood samples could be the simplest way to implement it. Nevertheless, its clinical utility has not yet been fully accepted. The reasons could be due to the rarity and heterogeneity of CTCs in blood samples that could lead to misleading results from assays only based on single CTC counts. To address this issue, a feasible direction is to combine the CTC counts with other clinical data for analysis. Recent studies have demonstrated the use of this new strategy for early detection and prognosis evaluation of cancers, or even for the distinguishment of cancers with different stages. Overall, this approach could pave a new path to improve the technical problems in the clinical applications of CTC counting techniques. In this review, the information relevant to CTCs, including their characteristics, clinical use of CTC counting, and technologies for CTC enrichment, were first introduced. This was followed by discussing the challenges and new perspectives of CTC counting techniques for clinical applications. Finally, the advantages and the recent progress in combining CTC counts with other clinical parameters for clinical applications have been discussed.

## 1. Introduction

Cancer has been a leading threat to global health in recent decades. According to World Health Organization (WHO) statistics, the number of people dying from cancer in 2018 was approximately 9.6 million [1]. In particular, 90% of cancer deaths are related to metastasis, highlighting the importance of early detection in the treatment of cancer [2]. Indeed, the survival rate for almost all cancers significantly improves when they are identified, diagnosed, and treated in their early stages [3,4]. The occurrence of cancer involves the imbalance of many complex molecular mechanisms and regulatory pathways. Based on these pathogenic mechanisms, the progress of biomarkers and target drugs in recent years has, indeed, brought progress to the diagnosis and treatment of cancer [5]. Simultaneously, researchers have also been actively searching for optimal cancer biomarkers or combinations that can be used for early cancer diagnosis and subsequent development monitoring.

Circulating tumor cells (CTCs), as novel cancer biomarkers, were first described by Dr. Ashworth in 1869 [6], and they were found to exist in the blood circulation of cancer patients with distant metastases. Detecting CTCs in blood samples is valuable due to the morphological resemblance between CTCs and primary tumor cells [6]. It holds significant promise in assessing and predicting the status of the primary tumor, offering substantial potential for a wide range of clinical cancer applications [7,8]. Previous research has shown that tumor cells can be transmitted even in the early stages of tumor development [9], which reveals that CTCs could be used for early cancer detection. In other aspects of clinical applications, counting CTC numbers in cancer patients’ peripheral blood samples was also reported to provide valuable insights into cancer prognosis [10,11]. For example, a higher CTC count has been demonstrated to correlate with more advanced disease stages, a poorer prognosis, and more sites of metastasis [12]. Meanwhile, monitoring changes in CTC numbers over time can help clinicians track the trajectory of cancer progression during treatment and adjust treatment programs in a real-time manner [13]. Taken together, counting and monitoring CTC numbers holds high potential for early cancer detection, prognosis evaluation, and therapeutic response monitoring for cancer patients.

Even though the clinical applications of CTCs are expected, precise counting of CTCs from surrounding blood cells remains technically challenging, mainly due to the rarity of CTCs (i.e., 1–10 CTCs in a 10 mL cancer patient’s blood sample) and the lack of unique CTC markers. To tackle the hurdles, the Cell Search platform, as the first and only FDA-approved system, was developed for the isolation, identification, and counting of CTCs based on multi-marker staining (i.e., EpCAM+, CK+, CD45−, and DAPI+) [14,15]. Its application was reported to predict the cancer progression or death of patients with various solid cancers (e.g., metastatic prostate cancer [16], breast cancer [17], colorectal cancer [18], head and neck cancer [15], and pancreatic cancer [19]. With advancements in cell analysis technologies, however, researchers also found that there is an issue of heterogeneity in CTCs in terms of genetic mutations, surface markers, and biological properties [20]. Moreover, certain specific subpopulations of CTCs (e.g., EpCAM− CTCs or CTC clusters) have been shown to play a critical role in cancer metastasis and the development of therapy resistance [21]. Their detection and counting are also clinically meaningful regarding cancer treatment or care. Therefore, based on the facts mentioned above, the conventional CTC marker-staining scheme (i.e., EpCAM+, CK+, CD45−, and cell nucleus+) might not be able to identify all CTC subpopulations. This highlights the possibility that current CTC detection methods might not be able to provide precise enough information for cancer care.

To overcome the bottleneck coming across in recent CTC detection, the combination of conventional CTC counting with other clinical parameters (e.g., CTC clusters, tumor blood markers, tumor imaging, personal physiological parameters, medical history, and cancer screening tests) for analysis could provide more precise detection results for clinical applications. For instance, in early cancer detection, CTC counts can be combined with some blood tumor markers (e.g., CEA, CA125, CYFRA21, and SCC) for analysis to improve the accuracy of lung cancer detection and even differentiate malignant pulmonary nodules (MPNs) from benign pulmonary nodules (BPNs) [22]. In addition to blood tumor markers, combining CTC counts with tumor imaging data has also shown its value in early cancer detection. Whether ultrasound (US) or mammogram (MMG), it has been confirmed that the combination of tumor imaging data with conventional CTC counts can help clinicians rapidly detect breast cancer at an early stage [23]. Similarly, the combination of CTC counts with the results of standard cancer screening tests, such as the immunochemical fecal occult blood test (iFOBT), also has a high potential to identify patients with colorectal cancer at an early stage [24]. Apart from early cancer detection, the integration of CTC counts, and other cancer-related parameters, has been reported to improve the prognostic assessment of cancer patients. For example, the information from the combined analysis of CTC counts and CTC clusters can be used to predict breast cancer patient outcomes (e.g., cancer progression or cancer death) [25] or the recurrence probability in colorectal cancer patients [26]. In terms of lung cancer, moreover, the combination of CTC counts with computed tomography (CT) can more accurately distinguish lung cancer types with different invasive capabilities and provide patients with more accurate follow-up treatment [27]. However, compared with the diagnosis and prognosis assessment of cancer, the studies relevant to the combination of CTC counts and other parameters for monitoring cancer treatment response are relatively few. In this review, the background information relevant to CTCs, including the characteristics of CTCs, the clinical use of CTC counting, and the technologies for CTC enrichment, were first introduced. This was followed by the discussion of challenges and new perspectives on applying the current CTC counting techniques for clinical practice. More importantly, this review discusses the combination of CTC counting with the routinely used clinical data to provide patients with more appropriate and accurate diagnosis strategies when the technical challenges related to CTC research have not yet been solved. 

## 2. Characteristics of CTCs

CTCs are a population of cancer cells detached from the primary tumor which enter the bloodstream, and their existence is highly correlated with cancer metastasis [28]. In cancer metastasis, CTCs can spread through human blood circulation and potentially lead to new distant metastatic lesion formation when CTCs are trapped in the capillaries of organs or tissues [29]. Based on the abovementioned phenomenon, the detection of CTCs in blood samples from cancer patients was regarded as an indicator for monitoring the status of the primary tumor and evaluating its development. Although researchers first observed CTCs in the blood of metastatic cancer patients in 1869 [6], the significance and application of CTCs in clinical cancer diagnosis and treatment were not accepted until advances in cell enrichment and isolation techniques were made. In particular, CTCs existing in blood circulation could be easily sampled by a vein blood draw, offering a promising, minimally invasive “liquid biopsy” method for oncologists to monitor and evaluate the status of nonhematologic cancers as a feasible alternative to current highly invasive tissue biopsies [30,31,32] (Figure 1).

In addition, in clinical cancer research, many studies have demonstrated that CTC numbers in the blood samples of cancer patients could be a biomarker for evaluating cancer progression. For example, in a lung cancer study, the results revealed that the CTC numbers in a blood sample of a patient would gradually increase along with the patient’s cancer stage) [12]. Specifically, counts below 3 CTCs/mL blood sample tended to correspond to stage I cancer, 3–20 CTCs/mL corresponded to stage II or III cancer, and exceeding 20 CTCs/mL exhibited a high risk of malignancy and distant metastasis, typically corresponding to stage IV cancer [33]. On average, a patient with metastatic carcinoma typically has a range of 5 to 50 CTCs/7.5 mL of blood [34]. The detection of higher CTC numbers might indicate a more aggressive disease state and a high recurrence rate for cancer patients [35,36]. In addition, as a clinical prognostic indicator, the detection of higher CTC numbers has been reported to be highly related to poorer survival rates of cancer patients in various solid cancers, such as breast, lung, prostate, and colorectal cancers [18,37,38,39]. For example, in metastatic breast cancer patients, it was found that the detection of more than 5 CTCs in 7.5 mL blood samples resulted in a shorter overall survival [17]. Moreover, according to clinical observations, the total amount of CTCs in the blood sample of cancer patients also decreased significantly after the administration of effective treatment, indicating that monitoring the change in CTC numbers can be an indicator for physicians to evaluate the cancer patient’s response to therapy [37]. Based on the abovementioned studies, counting and monitoring of CTC numbers in a specific amount of blood samples are commonly accepted as one of the feasible clinical CTC applications, especially the Cell Search system, whose mechanism of CTC counting has been FDA-approved for utilization in cancer patients’ prognosis evaluation [11].

In addition to the CTC counts, the analysis of CTCs’ characteristics also plays a pivotal role in a more comprehensive understanding of cancer biology, behavior, metastasis, and drug resistance, which can bring further applications for CTC detection [40]. Generally, owing to the heterogeneity of the primary tumor, CTCs, originating from the primary tumor, also exhibit remarkable cell heterogeneity in their genotypes, phenotypes, and morphologies [41]. For instance, Gasch et al. conducted a study that revealed that CTCs presenting heterogeneous *PI3K* mutations and *HER2* expression were detected in metastatic breast cancer patients [42]. Furthermore, the dynamic interplay between CTCs and their microenvironments is crucial. A tumor can be viewed as an integrated ecosystem where the co-evolution of neoplastic cells within the tumor microenvironment results in a wide range of cancer cell phenotypes. This is closely related to the tumor heterogeneity of CTCs [43]. Therefore, in a study on CTC morphologies, researchers found that CTCs in blood samples had a diverse array of sizes (diameter: 12–30 μm) [44] and shapes (e.g., clusters), reflecting that cancer cells maintained their high phenotypic plasticity in the bloodstream [45]. On the other hand, detection of CTC heterogeneity also reflects cancer progression, metastasis, and adaptation to environmental changes (e.g., chemotherapeutic resistance) [46]. For example, many studies found that an aggregate consisting of two or more CTCs (called a CTC cluster) prolonged the cell survival status in the bloodstream, enhanced the probability of cancer metastasis, and led to poorer prognosis in cancer patients than a single CTC [47,48]. Moreover, epithelial-mesenchymal transition (EMT) and mesenchymal-epithelial transition (MET) serve as two critical mechanisms in cancer metastasis [49,50], also affecting the phenotypic changes of CTCs. In the EMT and MET process, the CTC phenotypes dynamically change among epithelial CTCs (E-CTCs), mesenchymal CTCs (M-CTCs), and mixed CTCs (E/M-CTCs) [51]. CTCs in early-stage cancer patients may retain the characteristics of primary tumors and tend to be more epithelial. However, in advanced-stage cancer patients, due to the cell metastasis mechanism of cells undergoing the EMT process, the pattern of CTCs becomes more mesenchymal type [52]. This phenotypic plasticity based on EMT and MET enables CTCs to adapt to different metastasis stages and increases CTCs’ drug resistance, viability, motility, and invasion abilities [49,50]. However, even though a deep understanding of the cell characteristics and heterogeneity of CTCs is essential for developing new cancer diagnosis and treatment approaches, these detection techniques are commonly costly, time consuming, and complicated to implement [53,54]. In terms of current clinical CTC applications, therefore, counting CTC numbers and monitoring their dynamic changes have become the current mainstream options for cancer early detection, prognosis evaluation, and therapeutic response monitoring owing to their relatively low cost and simple detection process [55,56].

## 3. Clinical Use of CTC Enumeration/Counting

As aforementioned, counting and monitoring CTC numbers is a valuable and easy-to-use indicator for cancer status evaluations. In particular, the clinical utility of CTC counting has been widely investigated in various solid cancers, including lung, prostate, breast, colorectal, head and neck, and pancreatic cancers. Although current medical guidelines have not yet entirely accepted CTC-related applications, several academic studies have emphasized the potential of CTC counting in clinical use [57,58]. In this section, the clinical uses of CTC counting in cancer early detection, prognosis evaluation, and therapeutic response monitoring were discussed, which are also summarized in Table 1: CTC counts as a biomarker for clinical applications.

### 3.1. Early Detection of Cancer 

It is a well-known fact that 90% of cancer deaths originate from advanced cancer metastasis [2], which makes early cancer detection and timely treatment administration the key to decrease the cancer mortality rate [3]. And the presence of CTC in cancer patient at the early stages of cancer has been demonstrated in many studies [59,60], enabling CTC numbers to serve as an indicator to distinguish cancer patients and other cases (e.g., healthy volunteers or patients with benign diseases). For example, in breast cancer, Jin et al. indicated that setting the cutoff as 2 CTCs/4 mL of blood could significantly differentiate patients with stage 0–IV breast cancer from healthy volunteers and patients with benign breast diseases (AUC: 0.86) [59]. Similarly, appropriate cutoffs of CTC count for the distinguishment of cancer patients and other cases, including healthy or benign diseases, have been explored in the studies of lung cancer [60,61], colorectal cancer [62], and pancreatic cancer [63]. Even though many studies have demonstrated the potential of CTC counting in early cancer detection, its clinical use remains controversial. In one study, for example, the CTC-positive detection rate was measured to be only 50% and 80.43% in stage 0 and I breast cancer, respectively, compared to 87.5–100% in stage II–IV breast cancer [59]. Brown et al. also revealed a similar issue, showing that CTC counting may not be effective in early cancer detection due to its relatively low abundance, making it difficult to detect [64]. The abovementioned false-negative results of CTC detection in early-stage cancer could limit the application of using only CTC counts as an indicator in early cancer detection.

### 3.2. Prognosis Evaluation of Cancer

As cancer prognostic factors, higher CTC counts have been observed to correlate closely with unfavorable outcomes (e.g., cancer metastasis, therapeutic failure, or cancer recurrence) in various cancers, such as breast, colorectal, lung, and pancreatic cancer [12,19,37,65]. In these clinical studies, CTC numbers were set as an indicator to evaluate the probability of cancer patients recovering from or alleviating the disease or relapsing by monitoring survival duration until their death (i.e., OS: overall survival), cancer progression (i.e., PFS: progression-free survival), or cancer recurrence (i.e., RFS: recurrence-free survival or DFS: disease-free survival) [10,11,65]. For instance, in a study of hormone receptor-positive (HR+) metastatic breast cancer, the results showed that the OS and PFS of patients with ≥5 CTCs/7.5 mL after treatment were significantly worse than that of patients with <5 CTCs [66]. Similarly, in other clinical studies, researchers also observed that higher CTC counts detected in a blood sample were associated with worse OS or PFS in cancer patients with breast cancer [11,25], lung cancer [12,67], colorectal cancer [18,68], and pancreatic cancer [19,36]. Furthermore, in one small-cell lung cancer study, high CTC counts (≥ 10 CTCs/5 mL blood) were closely associated with advanced TNM stage (high lymph node metastasis and distant metastasis), indicating a more unfavorable prognosis. These results showed that CTC counts hold significant prognostic value for small-cell lung cancer patients (AUC > 0.50) [12]. In addition, high preoperative CTC counts were commonly detected in patients with cancer recurrence after surgical resection [35,36]. In another study involving pancreatic ductal adenocarcinoma, 75% of patients with preoperative CTC positivity (≥ 1 CTC/7.5 mL blood) relapsed within 12 months after surgical resection in comparison to 36.5% of CTC-negative patients [36]. Notably, 88.9% of preoperative CTC-positive and cancer recurrence patients were classified as having systemic recurrence, suggesting that preoperative counting of CTCs can predict early and systemic recurrence [36]. In brief, the prognostic value of CTCs was supported by the growing clinical data, which holds promise for clinicians to evaluate and adjust the treatment approaches in patients with high CTC counts [69]. Given that the Cell Search system is the first FDA-approved CTC counting device for cancer prognosis [57], this prognostic tool became the mainstream research and application direction of CTCs. 

**Table 1 cancers-15-05372-t001:** CTC counts as a biomarker for clinical applications.

Application	Cancer	Technique	Cutoff Value (CTCs/mL Blood)	Performance	References
**Cancer Diagnosis**	Breast cancer	CytoSorter^®^ microfluidic platform	≥2 CTCs/4 mL	AUC = 0.86	[59]
	Lung cancer	Cytelligen CTC Enrichment Kit and Human Tumor Cell Identification Kit	≥6 CTCs/6 mL	AUC = 0.780	[61]
		CellCollector + immunocytochemical staining	≥1 CTCs	AUC = 0.715	[60]
	Colorectal cancer	Negative immunomagnetic selection + Flow cytometry	≥3 CTCs/7 mL	AUC = 0.664	[62]
	Pancreatic cancer	The NE-imFISH enrichment system + fluorescence microscope	≥2 CTCs/3.2 mL	AUC = 0.85	[63]
**Prognosis evaluation of Cancer**	Breast cancer	Cell search	≥5 CTCs/7.5 mL	PFS: HR 1.79; OS: HR 2.72	[66]
		Cell search	≥5 CTCs/7.5 mL	DFS: HR 1.82; DDFS: HR 1.89; OS: HR 1.97	[11]
		CytoSorter^®^ microfluidic platform	≥5 CTCs/7.5 mL	PFS: HR 2.11; OS: HR 3.15	[25]
	Lung cancer	Negative immunomagnetic selection + FISH	≥10 CTCs/5 mL	Highly CTC is positively correlated with TNM stage and poor prognosis	[12]
		Cell Search	≥5 CTCs/7.5 mL	Poor PSF (4.1 months) and OS (4.6 months).	[67]
	Colorectal cancer	Cell search	≥1 CTCs/7.5 mL	CTC-positive patients showed distant metastasis and shorter PFS and OS than CTC-negative patients.	[18]
		Cell search	≥3 CTCs/7.5 mL	CTC-positive patients showed shorter PFS and OS than CTC-negative patients.	[68]
	Pancreatic cancer	MACS	≥1 CTCs/7.5 mL	CTC-positive patients had over four times shorter PFS and over two times shorter OS than CTC-negative patients.	[19]
		Ficoll-Paque PLUS	≥1 CTC/7.5 mL	Early recurrence: OR 8.770; systemic recurrence: OR 5.600	[36]
**Therapeutic Response Monitoring of Cancer**	Breast cancer	Cell search	≥5 CTCs/7.5 mL	After chemotherapy, the CTC positivity rate decreased. (T0:31% to T3:15%)	[66]
	Lung cancer	Cell search	≥1 CTC/7.5 mL	During immunotherapy, patients whose CTCs did not decline had poorer outcomes of the primary tumor.	[56]
		EasySep + Flow cytometry	≥3 CTCs/mL	After surgery, CTCs decreased, and the early rebound of CTC counts was positively associated with recurrence.	[70]
	Colorectal cancer	Cell search	≥2 CTCs/7.5 mL	During chemotherapy, the number of CTCs reflects the progression of the primary tumor.	[13]
		Immunofluorescence staining + RT-qPCR	≥3 CTCs/8 mL	After surgery, CTC appeared in stage T4 (nine months later) with local recurrence.	[71]

AUC, area under cure; PFS, progression-free survival; RFS, recurrence-free survival; DFS, disease-free survival; OS, overall survival; OR, odds ratio; HR, hazard ratio; NE-imFISH, negative enrichment and immune fluorescence in situ hybridization; MACS, magnetic activated cell sorting.

### 3.3. Therapeutic Response Monitoring of Cancer

Because CTC numbers are highly correlated with primary tumor progression, monitoring the dynamic changes in CTCs is highly valuable in assessing the therapeutic response of primary tumors [66,72,73]. Monitoring a decrease in CTC numbers during therapy indicates a good outcome; commonly direct tumor regression or disease control [13,67]. Conversely, an increase in CTC numbers represents a poor outcome to treatment. This could lead to therapeutic failure or progression of primary tumors, suggesting the need for further adjustments in therapy [37]. For example, in the study carried out by Mark et al., CTCs were counted at four time points (T0, T1, T2, T3) before and during treatment in HR+ metastatic breast cancer patients [66]. In these patients’ follow-ups, the number of CTCs (i.e., ≥5 CTCs/7.5 mL of blood) gradually decreased after the administration of the therapy (T0: 31.3% to T3: 15%) [66]. Meanwhile, the patients remaining CTC-positive at the four time points or becoming CTC-positive at T1 all showed a significant reduction in PFS or OS in comparison with the cases remaining CTC-negative. This study demonstrated that monitoring the change in CTC numbers could predict the outcome of cancer patients’ therapeutic response [66]. In another advanced non-small-cell lung cancer study, Menno et al. counted CTC numbers before and at 4–6 weeks of PD-L1 or PD-1 receptor inhibitor therapy [56]. The results showed that patients with increased CTC numbers during treatment had poorer outcomes in the primary tumor (e.g., cancer progression) than the cases with either CTC-negative or decreased CTCs at both time points [56]. Moreover, through several CTC counts of an advanced colorectal cancer patient during chemotherapy, Kazunori et al. also found that the decline and rise of CTC numbers were highly correlated with the good and poor outcomes of the patient’s therapy, respectively [13]. Notably, when chemotherapy fails to suppress the progression of a patient’s primary tumor, increased CTC numbers are detected earlier than blood tumor markers (i.e., CEA and CA19-9), which begin to increase [13]. Minimal residual disease (MRD) is a disease correlating to small numbers of cancer cells (e.g., CTCs) remaining cancer patients after curative treatment, which was demonstrated to be associated with the possibility of cancer recurrence [74]. As trace amounts of cancer cells that exist in the human blood circulation, monitoring changes in the number of CTCs after treatment (e.g., postoperatively) in patients can be used as an indicator for subsequent MRD or cancer recurrence [75]. Wu et al. counted the number of CTCs before surgery (day 0) and postoperatively (day 1 and day 3) in lung cancer patients. The results showed that CTC numbers significantly decreased in all patients after surgery. However, some cases emerged with an early resurgence of CTC counts on postoperative days 1 and 3, which subsequently correlated with recurrence occurring months later. The study demonstrates a potential clinical use of monitoring CTCs in detecting recurrence in early-stage lung cancer patients undergoing curative surgery [70]. Moreover, Hendricks et al. also found a significant increase in the CTC numbers detected at T4 (9 months) after surgery (0 cells at T0–T3) in colorectal carcinoma patients, and a continuous increase was observed at T5 (12 months). More importantly, local recurrence of the primary adenocarcinoma was detected, even though the tumor markers CEA and CA19-9 remained below their designated cutoff levels throughout the study [71]. These findings demonstrate that the accuracy of monitoring dynamic changes in CTC numbers could be more sensitive than traditional detection based on tumor markers [71]. As a whole, the surveillance of CTC numbers of alterations during cancer treatment can be an indicator to help evaluate the primary tumor response to therapies. 

## 4. The Technologies for CTC Enrichment 

Although clinical studies have supported the potential use of CTC counts in clinical cancer applications, the precise counting of all CTCs in a blood sample is still technically challenging. The main technical hurdles come from two issues: the rarity of CTCs in the blood samples (i.e., 1–10 CTCs in a 10 mL blood sample of a cancer patient) and the lack of unique CTC markers [76]. To address these issues, the general method for CTC counting mainly consists of three steps: (1) CTC enrichment, (2) CTC identification, and (3) CTC counting [77]. After acquiring enough blood samples from a cancer patient by vein blood draw, briefly, the first step (i.e., CTC enrichment) is performed to increase the CTC concentration in the sample either via the direct capture of CTCs (i.e., referred to as the positive selection of CTCs) or indirectly via the removal of the non-target blood cells, such as red blood cells (RBCs), white blood cells (WBCs), and platelets, in the sample (i.e., referred to as the negative selection of CTCs) [77]. In the following CTC identification step, multiple markers, such as the positive tumor-specific markers (e.g., commonly EpCAM or CKs), the cell nucleus markers (e.g., DAPI or Hoechst), and the negative blood cell-specific markers (e.g., CD45 of WBCs) [14,15] are utilized to distinguish the target CTCs from the surrounding blood cells. The CTC identification step is generally implemented via immunofluorescence staining, which could involve the use of fluorescence-observation or a detection device (e.g., fluorescence microscopy, fluorescence spectrophotometry, flow cytometry) [78] in the subsequent counting step (i.e., the third step). For further analysis of CTC characteristics (e.g., molecular biology-based assay or CTC culture), the CTCs could be further purified and isolated from the processed sample after the CTC identification or counting steps [79,80]. After the general description of the processes for the quantification of CTC numbers, this section aims to discuss the advantages and disadvantages of the current techniques used for CTC enrichment (i.e., the first step). Briefly, the current CTC enrichment techniques can be generally classified into two mechanisms: (i) based on the biophysical properties of CTCs (e.g., cell size, density, and bioelectrical properties) and (ii) based on the immune properties of CTCs (e.g., positive or negative immunoselection) [77]. Table 2 summarizes the current techniques for CTC enrichment. Table 2: Outlining circulating tumor cell (CTC) enrichment techniques based on various mechanisms 

### 4.1. Cell Biophysical-Based Techniques

For biophysical-based mechanisms, many studies have revealed that CTCs have several distinctive biophysical features, including a higher nucleus-to-cytoplasm ratio, larger sizes, distinct nuclear morphology, and unique electrical characteristics compared to normal blood cells [81]. This fact allows CTCs to be enriched based on the physical property differences between the CTCs and blood cells, as mentioned above, without any labeling [82]. For example, size-based CTC enrichment techniques mainly utilize the size differences between CTCs (diameter: 12–30 μm) [44] and blood cells (diameter of WBCs: 5–25 μm, RBCs: 6–9 μm, and platelets: 2–4 μm) [83,84] for separation purpose. Filter systems such as ISET [85] and Screencell [86] use membranes with specific pore sizes (e.g., 8 or 6.5–7.5 μm) to selectively capture CTCs from saponin-treated blood samples based on their large sizes. The Vortex VTX-1 system integrates microfluidics technology and laminar microvortices to trap large CTCs in its microfluidic chip [87]. In addition, owing to the significant density differences between mononuclear cells (e.g., CTCs, lymphocytes, and monocytes, density of <1.077 g/mL) and the cells with higher density (e.g., RBCs and polymorphonuclear leukocytes, density of >1.077 g/mL), density gradient centrifugation can serve as a simple and rapid approach to enrich CTCs in the first step [83]. For example, Ficoll-Hypaque^®^ uses a high-molecular-weight sucrose polymer as media to separate CTCs in a density gradient-based manner. However, the CTC recovery rate of Ficoll-Hypaque^®^ was reported to be low [83,88]. As an approach to improve it, OncoQuick^®^ integrates density gradient centrifugation and a filtration barrier in a 50 mL centrifuge tube to achieve a higher CTC recovery rate [88]. Moreover, dielectrophoresis (DEP) is a microscale particle manipulation technology that can be used to separate microparticles based on their size-induced dielectric differences [89]. Because CTCs are commonly larger than blood cells, some DEP-based techniques such as Apocell [90] or optically induced-dielectrophoresis (ODEP) systems [91,92] have also been developed to separate CTCs from surrounding blood cells based on size differences. As a whole, these cell biophysical-based CTC enrichment techniques are generally regarded to be simple to operate, labeling free, or high throughput. Nevertheless, these separation techniques are normally based on the differences in size or density between CTCs and other blood cells. In this situation, certain CTCs with similar sizes to the blood cells might not be effectively separated and enriched based on the current techniques. This could result in false-negative detection results. This limitation was confirmed by a breast cancer study revealing that the size of CTCs and WBCs could overlap [93]. Moreover, the achieved CTC purity based on the biophysical-based separation or enrichment techniques is commonly low [88], which could, in turn, cause problems in the subsequent CTC identification or counting. The general advantages and disadvantages of the cell biophysical-based techniques for CTC separation and enrichment are summarized in Table 2. Conversely, the CTC enrichment techniques based on the immune properties of CTCs could solve the technical problems coming across in the cell biophysical-based counterparts. These techniques are discussed in more detail in the following.

**Table 2 cancers-15-05372-t002:** (A): Outlining circulating tumor cells (CTCs) enrichment techniques based on various mechanisms (Cell biophysical-based techniques). (B) Outlining circulating tumor cells (CTCs) enrichment techniques based on various mechanisms (Immunoaffinity-based techniques).

**(A)**						
**Category**	**Typical Platform**	**Description**	**Capture Efficiency**	**Advantages**	**Disadvantages**	**References**
**Size-based**	ISET^®^	Filtration device	87%	Recovery of a heterogeneous population of CTCs, high purity, rapid, simple process	Sample loss during mononuclear cell depletion	[85]
	ScreenCell	Microfiltration device (pores of defined size for CTC capture)	74–91%	Recovery of a heterogeneous population of CTCs, high throughput, cheap and easy to produce, rapid process	Limited to larger CTCs, require additional enrichment step	[86]
	Vortex VTX-1	Inertial microfluidic chip (size-based separation)	53.8–71.6%	Recovery of a heterogeneous population of CTCs, fully automated process, high throughput (7.5 mL/20 min)	Limited to larger CTCs may require additional specific staining for cell identification.	[87]
**Density**	Ficoll-Hypaque	Blood is layered over a Ficoll-Hypaque	>90%	Simple and inexpensive	Low throughput (0.01−1.0 mL/h), high WBC contamination, loss of CTCs	[83,88]
	OncoQuick	Porous membrane filtration followed by density-grade centrifugation	84%	Simple and inexpensive	Loss of CTC, low purity	[88]
**Bioelectrical properties**	APOCELL	Dielectrophoresis field flow fractionation (size-based separation)	75%	Recovery of a heterogeneous population of CTCs, high throughput (7.5–10 mL/h), cell high viability, high purity	Require electric field frequency	[90]
	ODEP	Optically induced dielectrophoretic force-based cell manipulation in a microfluidic system (size-based separation)	63.6–85.6%	Recovery of a heterogeneous population of CTCs, label-free, high cell viability, high purity	Specific cell types and specific parameters require electric field frequency.	[91]
**(B)**						
**Category**	**Typical Platform**	**Description**	**Capture Efficiency**	**Advantages**	**Disadvantages**	**References**
**Positive selection**	Cell Search^®^	EpCAM antibody-coated magnetic beads + immunostaining	42–90%	(FDA)-cleared CTC detection technique	Low sensitivity, loss of CTCs	[11,17]
	AdnaTest	Immunomagnetic beads (cocktail antibody) + PCR	60–80%	Providing molecular characteristics of CTCs	High contamination of WBCs	[94,95]
	MagSweeper	A rod coated with EpCAM antibody-labeled magnetic beads	60–70%	High purity, high throughput (9 mL/h)	Expensive, loss of CTCs	[96]
	Isoflux	EpCAM antibody-coated beads in a microfluidic chip	73–81%	High purity, high cell viability	Time-consuming	[97]
	MACS	Immunomagnetic bead separation	40–90%	Low-cost, technically simple	Time-consuming, high cell loss, low purity (around 50%)	[98]
	CTC-chip	Microfluidic chip of microposts conjugated with anti-EpCAM antibody	60–90%	High capture specificity, high purity	Low throughput (1 mL/h)	[99]
	Herringbone chip	EpCAM antibody-coated microfluidic chip + immunofluorescence microscopy	>90%	High throughput (4.8 mL/h)	Low purity of captured CTCs (around 14%)	[100]
**Negative selection**	RosetteSep	Blood cells depletion by pelleting the RBC-WBC crosslinked immunorosettes	40–62.5%	High specificity, recovery of a heterogeneous population of CTCs,	Time-consuming, exclusion of CTC-WBC clusters	[101]
	EasySep	Immunomagnetic beads and anti-CD45 antibodies for WBC removal	19–65%	High throughput (1–4 mL/h), simple operation, recovery of a heterogeneous population of CTCs,	Variable recovery, exclusion of CTC-WBC clusters	[102,103]
	CTC-iChip	Incorporates the unique-designed DLD (deterministic lateral displacement) microstructure arrays, inertial focusing, and MACS	>90%	High throughput (8 mL/h), recovery of a heterogeneous population of CTCs,	Low purity of captured CTCs (around 8%)	[103]

### 4.2. Immunoaffinity-Based Techniques

Immunoaffinity-based CTC enrichment techniques are pioneered as one of the earliest methods for capturing CTCs [104], and their principle is based on the utilization of specific antibodies to selectively bind to cell surface antigens for distinguishing target CTCs and other cells. CTCs express tumor-associated cell surface antigens (e.g., EpCAM and CKs) that set them apart from other blood cells (i.e., positive immunoselection) [105]. Blood mainly consists of RBCs, WBCs, and platelets, conversely, CTC enrichment can be achieved by removing the majority of blood cells to collect the CTCs that do not express any blood cell-specific antigens (i.e., negative immunoselection) [105]. In this section, positive or negative immunoselection schemes for CTC enrichment are discussed.

#### 4.2.1. Positive Immunoselection-Based Techniques

As previously mentioned, positive immunoselection-based techniques directly capture CTCs from a blood sample by using the capture antibodies that specifically target tumor-associated antigens (e.g., EpCAM and CK) expressed on the surface of CTCs [105]. Microfluidics-based and immunomagnetic bead-based approaches are two commonly used strategies for performing positive immunoselection-based CTC enrichment [106]. In microfluidics-based techniques, specific capture antibodies (e.g., anti-EpCAM antibodies) are immobilized on the surface of microchannels in a microfluidic chip, which enables CTCs to be trapped in a microfluidic chip via their interaction with the surface antigens on CTCs [99,100]. In these microfluidics-based techniques, enhancing the interaction between cells and capturing antibody-functionalized surfaces is the key to increasing CTC recovery performance [99,100]. For example, Nagrath et al. proposed a CTC-chip with many micropost arrays on the inner surface of the microchannel to maximize contact between CTCs and anti-EpCAM antibodies-coated micropost surfaces [99]. Similarly, the herringbone (HB) chip designed by Stott et al. could induce microvortices in a microchannel to increase the interaction between the cells and the surface coated with antibodies [100]. Different from the microfluidic-based techniques, the positive immunoselection of CTCs based on immunomagnetic beads-based techniques is the most commonly-used method, mainly due to its user-friendly operation process and the mature commercial products available on the market [106]. For example, Cell Search [11,17], AdnaTest [94,95], MagSweeper [96], Isoflux [97], and magnetic-activated cell separation (MACS) technologies [98] are all reported as immunomagnetic bead-based methods for the positive immunoselection of CTCs. In terms of a working principle, capture antibody-coated immunomagnetic beads are generally used to selectively bind with CTCs. This is followed by the separation and recovery of the immunomagnetic beads-bound CTCs via magnet manipulation [98]. Most notably, the Cell Search platform, the first and only CTC counting equipment for cancer prognosis approved by the FDA, was developed for the positive immunoselection of CTCs through anti-EpCAM antibody-coated immunomagnetic beads [11]. Even though the Cell Search platform is a milestone for clinical CTC applications, its clinical utility is challenged by the growing culmination of clinical studies. In particular, the CTC recovery rate and cancer detection rate claimed by some commercially available techniques are lower than those obtained in practical clinical cases [107,108,109]. The reason behind this could be due to the dynamic changes in CTC surface antigen expression in the EMT process [110]. For example, in patients with advanced metastatic disease, CTCs tend to reduce the expression of surface antigens of EpCAM and CKs [107,108]. This could pose a challenge for positive immunoselection-based techniques in accurately counting these aggressive and low EpCAM-expressing CTCs, which may contribute to the lower CTC recovery rate and, thus, false-negative results [111]. Even though positive immunoselection-based CTC enrichment techniques are regarded as the most used approaches for CTC enrichment, the recovery of only high EpCAM-expressing CTCs is a problem for further clinical applications.

#### 4.2.2. Negative Immunoselection-Based Techniques

Due to the technical limitations of the positive immunoselection-based methods, as mentioned above, the development of a protocol that can obtain the possible CTCs without depending on tumor-associated cell surface antigens, such as EpCAM or CKs, is crucial. To achieve this purpose, a negative immunoselection-based CTC enrichment technique was proposed in which the recovery of CTCs depends on the removal of the other unwanted blood cells [105]. In practical operation, the RosetteSep (Stemcell Technologies) CTC enrichment technique utilizes multiple antibodies to target various blood cell markers (e.g., CD2, CD16, CD19, CD36, CD38, CD45, CD66b, glycophorin A, and CD36 or CD56). Then, density gradient centrifugation removes unwanted blood cells by pelleting the RBC-WBC crosslinked immunorosettes. The method was reported to achieve a 62.5% cancer cell recovery rate for the blood samples spiked with ovarian or prostate cancer cells [101]. Moreover, it successfully detected CTCs in 90% of blood samples from patients with remote metastatic epithelial cancer and 76.9% of blood samples from patients with prostate cancer [101]. In addition, another negative immunoselection-based CTC enrichment technique integrates RBC removal approaches (e.g., density gradient centrifugation, filter, or RBC lysis reagent) and an EasySep CD45 depletion kit (containing immunomagnetic beads and anti-CD45 antibodies for WBC removal) to remove the unwanted blood cells. However, the exact recovery rate of this technology is open to debate [102,103]. Furthermore, microfluidic chips such as CTC-iChip [103] are also designed for negative immunoselection-based CTC enrichment. CTC-iChip incorporates uniquely designed deterministic lateral displacement (DLD) microstructure arrays, inertial focusing, and MACS stages to remove RBCs, platelets, and immunomagnetic bead-bound WBCs. It was reported to have successfully collected EpCAM-negative CTCs from the blood samples of breast and prostate cancer patients [112]. The negative immunoselection-based CTC enrichment technique holds the potential to recover all types of CTCs by removing other blood cells in a sample. However, in practical operation, the CTC purity and recovery still need to be improved due to WBC contamination and equipment-induced CTC loss [113]. On the other hand, even though these negative immunoselection-based techniques can collect CTCs without the use of tumor-specific antigens, the subsequent CTC identification step still depends on common tumor-specific antigens (e.g., EpCAM or CKs). In this situation, some of the information relevant to the low EpCAM-expressing CTCs could be lost. This issue is discussed in more detail in the next section. Overall, the advantages and disadvantages of the immunoaffinity-based CTC enrichment techniques, as mentioned above, are also described in Table 2.

### 4.3. Challenges and New Perspectives of the Current CTC Counting Techniques for Clinical Applications

With the growing doubts about the relationship between clinical cancer data and CTC counts, some potential problems in the current CTC counting process must be reconsidered. In the current techniques, the two main technical hurdles to overcome are the rare numbers of CTCs in blood samples and the lack of unique CTC markers [76]. For the former, numerous CTC enrichment techniques based on various mechanisms (i.e., biophysics, positive immunoselection, or negative immunoselection) have been proposed to improve the CTC capture efficiency, as exhibited in Table 2. Some of these CTC enrichment techniques have been proven to achieve high CTC capture efficiencies (e.g., ≥90% [100]), significantly improving the technical issue of CTC loss before the subsequent counting process. For the latter, however, the conventional markers for the identification of CTCs only encompass the conventionally defined CTCs (i.e., EpCAM+, CK+, CD45−, and cell nucleus+ CTCs) [14], which could, therefore, lead to misleading CTC counting results. As described previously, the expression of these commonly used tumor-specific antigens, such as EpCAM, is not stable on the CTC surface [114]. Due to the EMT mechanism, as the EpCAM expression on the CTC surface downregulates, it promotes these low- or non-EpCAM-expression CTC subtypes to display a higher metastatic or drug-resistance ability [21,114]. Therefore, the current CTC counting schemes could underestimate these “atypical” CTC subtypes. As another form of CTCs, CTC clusters are also reported to have a high correlation with cancer progression and poor outcomes, which implies that the detection of CTC clusters could be a useful indicator for cancer prognosis [115,116,117]. Taken together, the combination of the abovementioned information (i.e., the numbers of low- or non-EpCAM-expression CTC subtypes and CTC clusters) is regarded as a feasible direction to improve the performance of current CTC counting in terms of CTC detection rate [21,114] and prediction accuracy of cancer patient outcomes [114].

Although the combination of several CTC-related parameters, as mentioned above, could achieve more accurate prediction results for cancer diagnosis or prognosis, the increase in complexity and cost may hinder its real clinical use. To address this issue, another feasible direction is to combine the results of conventional CTC counting with the routinely used clinical data relevant to cancer detection or screening (e.g., tumor blood markers, tumor imaging data, physical examination data, medical history, or cancer screening tests) for analysis. This strategy could compensate for the disadvantages of the current CTC counting scheme, as mentioned previously, in a cost-effective manner [23,27]. In particular, these clinical data relevant to cancer detection or screening have already been regularly measured and recorded in clinical cancer evaluation [118,119]. Based on the strategy, several studies have demonstrated that the combination of data from the sources, as mentioned above, can enhance the cancer detection rate compared with the use of only single data [23,27,61]. In the following sections, the potential clinical parameters for the purposes mentioned above, and the recent studies involving the utilization of the new strategy for clinical cancer applications, are discussed.

## 5. The Advantage of Combining the CTC Counts with the Other Clinical Parameters for Analysis

### 5.1. Potential Clinical Parameters for the Combination with CTC Count

The potential clinical parameters that can be combined with the conventional CTC counts for enhancing their performance in cancer diagnosis, prognosis, and monitoring include the CTC-related parameters (e.g., the subtypes of CTCs or CTC clusters) and the other clinical parameters (e.g., tumor blood markers, tumor imaging data, patient’s physiological parameters, medical history, the results of cancer screening tests). All of these possible clinical parameters are summarized in Table 3 (Potential Clinical Parameters for the Combination with CTC counts) and described in the following.

#### 5.1.1. CTC-Related Parameters (e.g., Atypical CTC Subtypes or CTC Clusters)

Recent clinical studies have demonstrated that conventional CTC count does not precisely reflect cancer patients’ pathological differences because the information behind the subtypes of CTCs is generally ignored in current EpCAM-dependent CTC counting methods. Therefore, the clinical information hidden behind the subtype of CTCs has to be included in the analysis. Overall, CTCs can be classified into four types: typical and commonly used epithelial CTCs (E-CTCs), mesenchymal CTCs (M-CTCs), partial EMT CTCs (pEMT-CTCs), and stem cell-like CTCs (SC-CTCs) [51]. In order to distinguish these different subtypes of CTCs, in addition to the traditional EpCAM antibodies, different antibodies such as vimentin or CD44 can also be used to classify CTCs with specific targets [51]. Clinically, the number of subtypes of CTCs has been confirmed to be associated with different types and statuses of cancers [21,51], which was not reflected in the current counting of only typical E-CTCs. In addition to the subtypes of CTCs, CTCs have also been found to form cell clusters in the blood circulation of cancer patients via the aggregation of CTCs themselves (homotypic CTC clusters) or with immune cells such as white blood cells (heterotypic CTC clusters) [120]. Among them, CTCs that heterogeneously combine with neutrophils to form clusters through neutrophil extracellular traps can escape immune surveillance by blocking peripheral leukocytes activation [121]. As previously mentioned, CTC clusters in the bloodstream have been linked to advanced disease stages and poor clinical outcomes in several cancer types [115,122]. Regardless of the type of CTC cluster, their detection mainly relies on microscopy-related applications. Compared with the detection of single CTCs, it requires more antibodies for operation and reduces the physical damage caused during enrichment and isolation, making it more difficult to detect CTC clusters [123]. Although the clinical significance of these other CTC-related parameters has been indicated in some clinical research, the efficiency of the method used to isolate these CTC-related parameters needs to be further validated [115,122,124].

#### 5.1.2. Tumor Blood Markers

The detection of tumor-related markers in the bloodstream is the widely employed method for cancer diagnosis in clinical medicine [118]. Detecting these specific cancer-associated targets in a blood sample offers patients a minimally invasive, rapid, and easily accessible screening tool [125]. The selection of tumor blood markers varies according to the type of cancer being investigated. For instance, prostate-specific antigen (PSA) is utilized in prostate cancer diagnosis [126]. Thyroglobulin plays a crucial role in thyroid cancer detection, and cancer antigen 19-9 (CA 19-9) is employed in diagnosing pancreatic cancer, hepatocellular carcinoma, and gastric adenocarcinomas [127]. Carcinoembryonic antigen (CEA) is a crucial marker for colorectal cancer [128]. Ovarian cancer diagnosis relies on human epididymis protein 4 (HE4), CA125 is explicitly used in [129], and hepatocellular carcinoma is diagnosed using alpha-fetoprotein (AFP) [130]. As a whole, these tumor biomarkers have earned their place as integral components of patient management and are incorporated into various clinical guidelines for cancer diagnosis and treatment [131]. 

**Table 3 cancers-15-05372-t003:** Potential Clinical Parameters for the Combination with CTC counts.

		Parameter	References
**CTC-related**	Atypical CTC subtypes:	Epithelial (E-CTCs), mesenchymal (M-CTCs), and partial epithelial/mesenchymal (pEMT-CTCs), stem cell like (SC-CTCs).	[51]
	CTC clusters:	Homotypic CTC clusters, heterotypic CTC clusters (CTC-immune, CTC-stroma cells).	[120]
**Traditional medical testing (non-CTC-related)**	Tumor blood markers	Alpha fetal protein (AFP), carcinoembryonic antigen (CEA), carbohydrate antigen 19-9 (CA19-9), cytokeratin fragment 21-1 (CYFRA21-1), squamous cell carcinoma antigen (SCC), prostate-specific antigen (PSA), carbohydrate antigen 15-3 (CA15-3), carbohydrate antigen 125 (CA125), human epididymis secretory protein 4 (HE 4), lactate dehydrogenase (LDH), thyroglobulin, neuron-specific enolase (NSE), nuclear matrix protein 22, prostatic acid phosphatase (PAP)…	[126,127,128,129,130]
	Tumor imaging data	X-ray imaging, computed tomography (CT), positron emission tomography (PET), ultrasound sonography (US), and nuclear magnetic resonance imaging (MRI).	[132]
	Patient’s physiological parameters	General survey (height, weight, gender, age); vital signs (blood pressure, body temperature); skin (skin moisture, dryness, temperature, color…); neck (palpate the cervical lymph nodes); back (palpate the spine and muscles) and various systematic examinations of the individual system	[133,134]
	Medical history	Personal medical history, past surgical history, family medical history, and social history.	[135]
	Cancer screening test	Fecal occult blood test (FOBT), Cologuard, Pap smear	[136,137,138,139]

#### 5.1.3. Tumor Imaging Data

Tumor imaging data are crucial in clinical oncology for diagnosing and assessing cancer treatment strategies [140,141,142]. Various imaging techniques, including X-ray, computed tomography (CT), magnetic resonance imaging (MRI), and ultrasound (US), are routinely employed in cancer medicine [132]. These techniques provide tumor imaging data to enable physicians to evaluate cancer progression using imaging parameters such as the size of the primary tumor and the extent of tumor spread, whether locally or through distant metastasis. This information is useful in assessing the effectiveness of cancer treatment strategies [132]. 

#### 5.1.4. Patient’s Physiological Parameters

The patient’s physiological parameters serve as essential data for physicians to assess the patient’s overall health and medical condition [143]. These parameters can be obtained through physical examinations or collecting information on the patient’s lifestyle risks and past medication history, typically provided in healthcare records. In the case of cancer patients, these physiological parameters often exhibit varying patterns of change throughout cancer progression and treatment. This includes noteworthy deviations like fluctuations in weight and blood pressure. Such data can be shared with medical professionals, enabling them to accurately assess the patient’s condition and make necessary adjustments to their treatment plans [133,134].

#### 5.1.5. Medical History

Personal and family medication and medical history help in understanding a patient’s cancer risk and individual constitution [135]. Previous studies have pointed out that cancer is related to hereditary genes [144]. Physicians can also use the patient’s self-description to understand the risk of cancer exposure, such as smoking and drinking, and incorporate them into the parameters for the cancer patient’s risk management and assessment [145].

#### 5.1.6. Cancer Screening Test

Cancer screening tests are the tests that help detect cancer before significant symptoms appear to prevent advanced cancer and even death. The forms of cancer screening tests vary with the cancer types due to the different properties and sites of cancers. For example, screening for precursors of cervical cancer through Papanicolaou (Pap) smears has been demonstrated to significantly decrease the occurrence of invasive cancer on a global scale [139]. In addition, stool-based tests like the fecal occult blood test (FOBT), and fecal DNA testing (e.g., Cologuard^®^) are commonly used for early screening of colorectal cancer by detecting the bleeding of polyps or cancer during bowel movement [136,137,138]. These low-invasive cancer screening tests offer convenience, safety, and cost effectiveness, providing valuable early detection and intervention opportunities to improve patient outcomes and reduce cancer-related morbidity and mortality [146].

### 5.2. Clinical Utility of Combination CTCs with Multiple Parameters

Numerous clinical studies have revealed the clinical significance of CTCs in cancers. The number of CTCs in blood samples holds great promise in various clinical applications, including early cancer detection, prognosis assessment, and treatment response monitoring. However, the current CTC counting techniques are limited in providing more comprehensive information for the above clinical applications. To address this issue, several clinical research studies have explored the feasibility of combining data from the conventional CTC counts and other clinical parameters, as illustrated in Figure 2, to improve the performance of early cancer detection, prognosis evaluation, and other clinical uses. This innovative approach could pave a new path for solving the technical problems in the current CTC counting techniques and their clinical applications. In this section, the relevant studies in this new direction are summarized in Table 4 (The Combination of CTC Counts with Other Clinical Parameters: Its Performance and Clinical Utility in Cancer Management) and described in the following.

The analysis strategy of combining CTC count with other cancer-related parameters can be mainly divided into three steps. The first step is to isolate and enrich CTCs from the patient’s peripheral blood, and then detect and count them. The second step is to cooperate with medical staff to collect other parameters related to the patient’s cancer. Finally, through research and data analysis, the best combination of parameters can be found based on the detection goals (such as early diagnosis, treatment monitoring, and prognosis assessment), and more detailed clinical results can be obtained. Created with BioRender.com accessed on 3 October 2023. Abbreviations: CEA, carcinoembryonic antigen; CA125, carbohydrate antigen 125; CT, computed tomography; ultrasound sonography; FOBT, fecal occult blood test.

#### 5.2.1. Early Detection of Cancer

Early cancer detection is important in clinical cancer treatment. Although CTC counting is considered promising in early cancer detection, the rare number of CTCs in the early stage makes them technically challenging to detect. This phenomenon could, in turn, affect its performance in cancer detection. With the combination of other clinical parameters, this technical problem could be improved. For example, breast cancer detection is the most recognized clinical application of CTCs [59,60]. Even so, there are difficulties in using conventional CTC counting to detect early breast cancer. To address this issue, a retrospective case–control study combined CTC counts with multiple tumor blood markers (CEA, CA125, and CA15-3) to analyze the correlation in patients with breast cancer, benign breast diseases, and healthy female donors. The findings revealed that combining CTC counts with either CEA, CA125, or CA15-3 for diagnostic purposes yielded higher accuracy than using any single parameter alone. Notably, the most effective detection combination emerged from the comprehensive analysis of CTCs, CEA, and CA15-3 (AUC = 0.874). Overall, the combination of CTC counts with CA15-3 was indicated to be sufficient for achieving a favorable diagnostic performance in breast cancer (AUC = 0.873) [147]. As a result, combining CTCs with tumor blood markers can help to differentiate breast cancer patients at all stages from healthy individuals. In addition to breast cancer, the diagnostic accuracy of lung cancer is also an issue that urgently needs to be improved. Due to the poor prognosis of lung cancer and the high probability of lung cancer metastasis, this highlights the importance of detecting the occurrence of lung cancer at an early stage. For this purpose, Li et al. conducted a study utilizing the negative screening technology fluorescence in situ hybridization (NE-FISH) to isolate and count CTCs in the blood samples of lung cancer patients. They assessed the correlation between the number of CTCs and various lung cancer-related markers, including CEA, CA125, CYFRA 21-1, and SCC. The study yielded noteworthy findings: when CTC counts were integrated with other tumor blood markers, a significant enhancement in the sensitivity of lung cancer diagnosis was observed, rising from 68.29% to an impressive 82.93% [148]. More importantly, this combined analysis method can also distinguish between malignant pulmonary nodules (MPNs) and benign pulmonary nodules (BPNs) in lung tissue [22]. Moreover, pancreatic ductal adenocarcinoma (PDAC) patients are generally diagnosed at an advanced stage due to its deep location in the human body and detection challenges, resulting in a 5-year survival rate of 11% [149,150]. Therefore, the early diagnosis of PDAC is urgently needed. CA19-9 is a blood marker that has received approval from the FDA for the routine management of PDAC [151]. A comprehensive analysis evaluated the effectiveness of combining CA19-9 levels and CTC counts in the bloodstream for early PDAC diagnosis [63]. CA19-9 and CTC counts each have essential accuracy in the diagnosis of PDAC (AUC = 0.8 and 0.85, respectively). The combination of them for analysis can significantly increase the accuracy of diagnosing early PDAC (AUC = 0.95) [63]. 

**Table 4 cancers-15-05372-t004:** The combination of CTC counts with other clinical parameters: its performance and clinical utility in cancer management.

Application	Cancer	Aim	Combinations	Parameter	Cutoff Value	AUC/Outcome	References
Cancer diagnosis	Breast cancer	Distinguish breast cancer patients from benign and healthy participants	CTC + tumor blood markers	CTC	2 cells/mL blood	0.845	[147]
				CEA	>5 ng/ml	0.623	
				CA125	>35 U/ml	0.477	
				CA15-3	>25 U/ml	0.58	
				CTC + CEA		0.866	
				CTC + CA125		0.848	
				CTC + CA15-3		0.873	
				CTC + CEA + CA125		0.868	
				CTC + CEA + CA15-3		0.874	
				CTC + CA125 + CA15-3		0.873	
				CTC + CEA + CA125 + CA15-3		0.874	
		Early-stage (I-II) breast cancer diagnosis	CTC + tumor imaging data	CTC	2 cells/4 mL blood	0.855	[23]
				US (Ultrasound)	4 b	0.861	
				MMG (Mammogram)	4 b	0.759	
				CTC + US		0.922	
				CTC + MMG		0.899	
	Lung cancer	Early-stage (I-II) lung cancer diagnosis	CTC + tumor blood markers	CTC	2 cells/3.2 mL blood	0.825	[148]
				CEA	>5 ng/ml	0.541	
				CA125	>35 U/ml	0.565	
				CYFRA21-1	>3.3 ng/ml	0.587	
				SCC	>1.5 ng/ml	0.509	
				CEA + CA125 + CYFRA21-1+ SCC	>1.5 ng/ml	0.647	
				CTC + CEA + CA125 + CYFRA21-1 + SCC		0.854	
		Distinguish MPN patients from BPN patients		CTC	2 cells/3.2 mL blood	0.813	
				CEA	>5 ng/ml	0.546	[22]
				CA125	>35 U/ml	0.588	
				CYFRA21-1	>3.3 ng/ml	0.596	
				SCC	>1.5 ng/ml	0.551	
				CEA + CA125 + CYFRA21-1 + SCC		0.67	
				CTC + CEA + CA125 + CYFRA21-1 + SCC		0.853	
		Distinguish benign from pulmonary nodules <2 cm	CTC + tumor imaging data	CTC	6.05 cells/4 mL blood	0.843	[27]
				CT		0.83	
				CTC + CT		0.918	
		Lung cancer diagnosis in patients with SPNs	CTC + tumor blood markers + tumor imaging data	CTC	6 units/6 mL blood	0.78	[61]
				CEA	2.09 ng/ml	0.626	
				Size (mm) (CT imaging)	8 mm	0.572	
				NT (CT imaging)	−600 HU	0.626	
				Site (CT imaging)		0.555	
				CTC + CEA		0.734	
				CTC + CEA + NT		0.827	
				CTC + CEA + NT + Size + Site		0.841	
	Colore-ctal cancer	Colorectal cancer screening	CTC + tumor blood markers + cancer screening test	CTC	23 cells/mL blood	0.8602	[24]
				CEA	>5 ng/ml	Detection rate: 30.3%	
				iFOBT		The false-positive rate of iFOBT: 56.3%	
				CTC + CEA		Detection rate: 89.9%	
				CTC + iFOBT		Reduced false-positive rate of iFOBT to 18.8%	
	Pancre-atic cancer	Pancreatic ductal adenocarcinoma diagnosis	CTC + tumor blood markers	CTC	≥2 cells/3.2 mL blood	0.85	[63]
				CA19-9	≥37 U/ml	0.8	
				CTC + CA19-9		0.95	
Prognosis evaluation of the cancer	Breast cancer	Predict outcomes in metastatic breast cancer patients at baseline	CTC + CTC-related	CTC	≥5 cells/7.5 mL blood	PFS: HR 1.74 OS: HR 1.84	[25]
				CTC + CTC clusters	≥5 cells + ≥1 CTC-cluster/7.5 mL blood	PFS: HR 5.16 OS: HR 7.79	
	Colorectal cancer	Predict recurrence for patients at all colorectal cancer stages	CTC + CTC-related + tumor blood markers	CTC	>3 cells/2 mL blood	Recurrence rate: 32%	[26]
				CEA	> 5 ng/ml	Recurrence rate: 24%	
				CA19-9	> 37 ng/ml	Recurrence rate: 36.4%	
				CTC clusters	>0	Recurrence rate: 45.2%	
				CTCs + CEA		Recurrence rate: 45.5%	
				CTCs + CA19-9		Recurrence rate: 57.1%	
				CTCs With CTC clusters		Recurrence rate: 64.7%	
				CTCs + CEA + CA19-9 + With CTC clusters		Recurrence rate: 100%	
	Digestive tract cancer	Predict postoperative recurrence of cancer	CTC + tumor blood markers	CTC	6.87 cells/7.5 mL blood	0.831	[152]
				CEA mRNA	3816.20 copies/mL	0.912	
				CTC + CEA mRNA		0.965	
Others	Lung cancer	Tumor invasiveness prediction	CTC + tumor imaging data	FR + CTC Tumor volume (AI-assisted diagnosis system, ScrynPro) FR + CTC + tumor volume	9.75 FU/3 mL blood	0.659	[153]
					118 mm^3	0.698	
						0.841	
	Colorectal cancer	Metastasis prediction	CTC + CTC-related + tumor blood markers	CTC CEA CTC + CEA	≥3 cells/7 mL blood	0.664	[62]
					>5 ng/ml	0.78	
						0.837	

US, ultrasound; MS, Mammogram; CT, computed tomography; NT, Nodule type; HU, Hounsfield units; U, units; FR + CTC, folate-receptor-positive circulating tumor cell; FU, folate-receptor unit; AI, artificial intelligence.

Cancer screening techniques based on imaging systems are widely used in clinical practice. The frequently used methods include mammography (MMG), ultrasound (US), and magnetic resonance imaging (MRI). In addition to tumor blood markers, tumor imaging data can be combined with CTC counts as a matching option to enhance the performance of CTC counts-based early cancer detection. In breast cancer, for example, it was found that the diagnostic accuracy of early-stage breast cancer increased significantly compared with the individual use of US and MMG (AUCs increasing from 0.861 to 0.922 and 0.759 to 0.899, respectively) when they were combined with CTC counts. Further, considering the specificity issue, the study also pointed out that the best combination is the detection of CTC numbers and MMG. These findings highlight the potential of combining CTC analysis with the existing tumor imaging methods for more effective early-stage breast cancer screening [23]. In addition to breast cancer, the combination of tumor imaging data and CTC counts has also been found useful in lung cancer. Sequential computed tomography (CT) is commonly used for patients with indeterminate pulmonary nodules. The combination of CTC counting and CT has also been considered a diagnostic strategy that can increase accuracy in lung cancer [27]. In terms of the combination of CTC counts, tumor blood markers, and tumor imaging, a lung cancer study revealed that CTC-based lung cancer detection exhibited the highest accuracy when it was compared with those based on the other lung cancer-related parameters, such as CEA, nodule size, and nodule type (AUC = 0.780 > 0.626, 0.572, 0.626). Furthermore, its AUC increased to 0.841 when the CTC counts were combined with CEA, nodule type, nodule size, and nodule location for analysis. Overall, these findings demonstrate that CTC counts combined with multiple parameters for analysis can effectively detect early-stage lung cancer [61]. In addition to the abovementioned clinical parameters, the common references for cancer diagnosis include physical examination data, drug history, and cancer screening tests (e.g., iFOBT). For early detection of colorectal cancer, for example, iFOBT is a relatively simple method for the screening of colorectal cancer [136,137]. To enhance the performance of cancer prediction, Tsai et al. combined CTC count with iFOBT or CEA. The results showed that integrating iFOBT with CTC counts could significantly reduce the false-positive rate of iFOBT from 56.3% to 18.8–23.4%. When serum CEA was combined with CTC counts, the effectiveness of disease detection was substantially enhanced from 30.3% to 86.2–89.9% [24].

#### 5.2.2. Prognosis Evaluation of Cancer

Apart from the utilization of the combined data for CTC counts and other clinical parameters for the high-accuracy detection of cancers, this strategy can also be applied to improving the performance of cancer prognosis evaluation. Several studies have found that CTCs not only exist in single forms in the blood but also form clusters in plural forms. CTC clusters were reported to have distinct characteristics and play a significant role in metastasis [154,155]. In addition to the aforementioned clinical parameters, as previously mentioned, the combination of CTC counts, and CTC cluster counts for analysis is useful for the prognosis evaluation of metastatic breast cancers. It was found that the OS and PFS of patients in the high-risk group with more CTCs and CTC clusters in the blood were significantly lower than those of other groups. Moreover, the disease progression and number of deaths associated with breast cancer were also considerably more severe in this group [25]. In addition, the analysis of CEA mRNA in peripheral blood and resected lymphoid tissue is a specific detection method that helps to detect tumor micro-metastasis [156]. In digestive tract cancer, a study was performed to investigate whether CTC counts and CEA mRNA can predict the probability of recurrence in patients who underwent radical resection. According to the analysis, the CTC numbers and CEA mRNA levels in the blood of relapsed patients were significantly higher than those of nonrelapsed patients. After correlation analysis, the effectiveness of using the two parameters in combination was significantly higher than using CTC count alone (AUC = 0.965 > 0.831) for recurrence prediction [152]. Furthermore, clinical researchers also try to use the combination of CTC clusters, CTC counts, and tumor blood markers for analysis to predict the recurrence rate of colorectal cancer patients [26]. When the recurrence rate was evaluated alone by the number of CTCs, CEA, CA19-9, and CTC clusters, only patients with CTC clusters had a higher recurrence rate (45.2%). It can also be found that the patient recurrence rate is the highest (64.7%) under the condition of high CTC counts and the occurrence of CTC clusters. Notably, when all these parameters are included in the evaluation, the recurrence rate of patients is as high as 100% [26]. On the whole, the combination of CTC counting with other clinical parameters for analysis can be helpful in the prognosis assessment of cancer patients.

#### 5.2.3. Others

In addition to cancer diagnosis and prognosis assessment, distinguishing cancers with different stages and pathological features is also a clinically necessary project, especially in cancer metastasis, which is strongly related to patient mortality. The status of cancer metastasis will directly affect the patient’s cancer stage, and the treatment strategy will also be different accordingly. Many cancer staging systems, such as TNM and Barcelona Clinic Liver Cancer (BCLC), refer to the patient’s metastasis status for reference assessment and provide guidelines for treatment strategies [157]. Considering that the combination of CTC counts and other clinical parameters for analysis could improve the performance of cancer diagnosis and prognosis evaluation, this strategy could be feasible for the improved assessment of cancer metastasis. In colorectal cancers, combining CTC counts with CEA for analysis has demonstrated its clinical utility in both diagnosis and prognosis. Based on this, its ability to differentiate metastatic colorectal cancer patients was explored. Not surprisingly, it was found that after the joint analysis of CTC counts and CEA, the ability to assess the presence or absence of metastasis in colorectal cancer patients increased (AUC = 0.837) [62]. Moreover, it is well recognized that different pathological histological types can affect lung cancer patient’s prognosis and treatment strategies. These different subtypes have various prognostic assessments and treatment strategies. In the past, different subtypes of lung cancer mainly relied on CT to evaluate the volume of pulmonary nodules or other parameters related to invasiveness based on artificial intelligence (AI) assistance [158]. In recent years, it has been pointed out that evaluating folate receptor (FR)+ CTCs has good application value in breast and lung cancer [159]. Besides, the studies have indicated synergy between the CTC counts and tumor volume in small-cell lung cancer. Based on these past experiences, an attempt was made to combine the number of FR + CTCs with the AI-assisted diagnosis tumor volume system (ScrynPro) to predict lung cancers with different invasive abilities. After the analysis, compared with using FR + CTCs and tumor volume alone, the combination of the two can be used to predict the invasion ability of lung cancer more accurately (AUC = 0.841) [153].

### 5.3. Challenges and New Perspectives 

In this review, we have discussed the technical advantages of combining the CTC counts with other conventional cancer-related parameters for analysis to overcome the limitation of the current CTC counting method. These combinations have been successfully demonstrated to improve the accuracy in early detection and prognosis evaluation of cancers. To obtain these clinical data for analysis, however, it could, to some extent, increase the economic burden for patients and medical institutions. In addition, an inappropriate combination of the clinical data might not be able to enhance the accuracy of cancer-relevant evaluation. Finding the appropriate combinations of clinical data, therefore, would be an important issue. This issue could be explored by effective data mining of big data, or with the aid of artificial intelligence (AI) and machine learning. Apart from the combination of other conventional cancer-related parameters for analysis, the data relevant to the CTC subtypes could provide clinically valuable information. To obtain this information, more sophisticated analytic equipment like imaging flow cytometry is required. Moreover, with the application of aptamers, research on designing different aptamers for the purpose of more specific and comprehensive CTC detection is also increasing. The application of DNA or RNA aptamers has already produced impressive results in lung and breast cancer [160,161]. Especially in the detection of CTCs in non-epithelial cell cancers such as glial brain tumors, the application of aptamers also has considerable potential [162]. Therefore, even though many challenges exist, the com-prehensive application of CTCs in cancer clinics is still expected to be achieved.

## 6. Conclusions

CTCs are a population of cancer cells detached from the primary tumor and enter the bloodstream. Their existence in a blood sample is highly correlated with cancer metastasis. Therefore, the information hidden behind these cells holds great promise for early cancer detection, prognosis evaluation, and even therapeutic response monitoring of cancer. The acquirement of clinically valuable information from CTCs can be achieved through the analysis of CTCs’ characteristics. Although more comprehensive information on CTCs could enhance the performance of clinical detection or prediction work, the assays relevant to this work are normally costly, time-consuming, and complicated to implement. This could hinder their widespread clinical applications. Conversely, the CTC numbers in the blood samples have been clinically proven to be a useful biomarker for early cancer detection, prognosis evaluation of cancer, and therapeutic response monitoring of cancer. With recent advances, the techniques for CTC counting have become mature. Compared with the analysis of CTCs’ characteristics, therefore, CTC counting is more feasible to practically implement. Nevertheless, its real clinical utility has not yet been fully accepted. The main reasons behind this could be due to the rarity of CTCs in blood samples and the heterogeneity of CTCs (i.e., the CTC subtype issue) that could lead to misleading results of the assays only based on single CTC counts. Considering the performance of evaluation and the issue of practical application, a feasible direction is to combine the results of conventional CTC counting with the routinely used clinical data relevant to cancer detection or screening (e.g., tumor blood markers, tumor imaging data, physical examination data, medical history, or cancer screening tests) for analysis. This strategy could compensate for the limitations of the current CTC counting scheme in a cost-effective manner. Within the cases discussed in this review, overall, the combination of CTC counts with other clinical parameters for analysis has been successfully demonstrated for the early detection of breast, lung, colorectal, or pancreatic cancer with enhanced performance compared with those based on the use of only CTC counts. Apart from the high accuracy detection of cancers, this strategy can also be applied to improving the performance of cancer prognosis evaluation in metastatic breast, digestive tract, or colorectal cancer. Furthermore, this new strategy is also found useful for the assessment of the presence or absence of metastasis in colorectal cancer patients or for the prediction of lung cancers with different invasive abilities with enhanced performance. On the whole, this innovative approach could pave a new path to improve the technical problems coming across in the current CTC counting techniques and their clinical applications. 

## Figures and Tables

**Figure 1 cancers-15-05372-f001:**
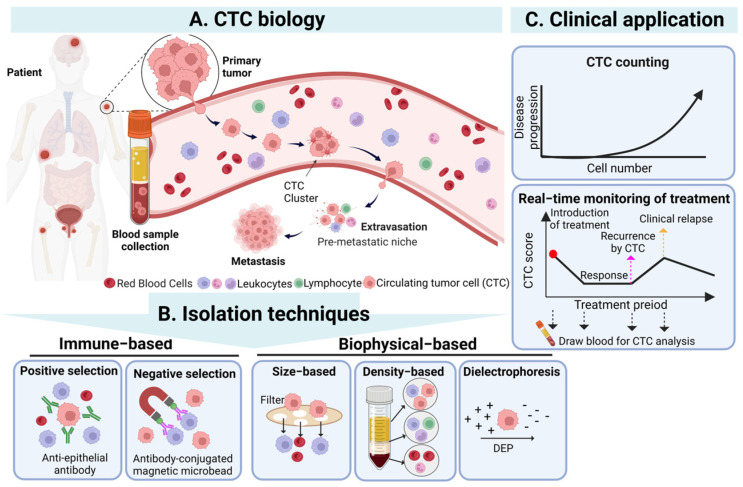
The overview of CTCs’ biology, isolation techniques, and clinical application of CTC. (**A**) CTCs detach from the primary tumor, enter the blood, and then undergo the stages of intravasation, circulation, and extravasation to reach the occurrence of distant metastasis of the tumor. (**B**) CTCs in the blood can be isolated through immunological and biophysical principles. Immunological methods include positive and negative screening, while biophysical methods include based-on-size, density, and dielectrophoresis. (**C**) The isolated CTCs can be used clinically as a biomarker and an effective tool for diagnosis, prognosis, and treatment monitoring. Created with Biorender. Abbreviations: DEP, dielectrophoresis.

**Figure 2 cancers-15-05372-f002:**
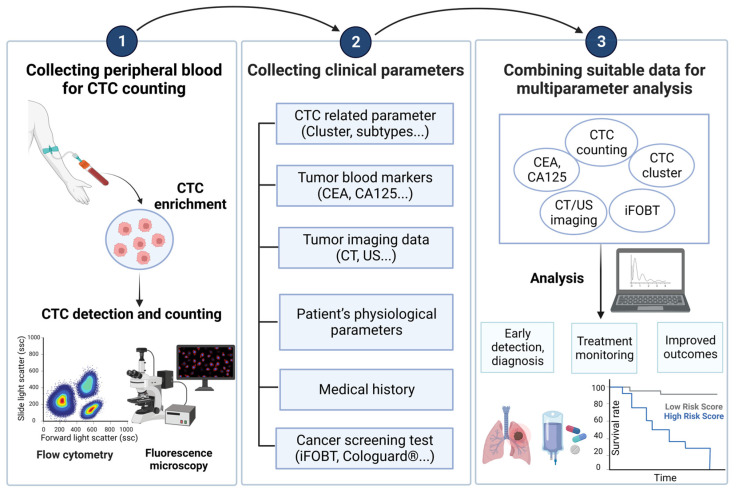
A brief description of the combined analysis of CTC counts and cancer-related parameters in clinical practice.

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
