# Peer review of "Recent Progress in Enhanced Cancer Diagnosis, Prognosis, and Monitoring Using a Combined Analysis of the Number of Circulating Tumor Cells (CTCs) and Other Clinical Parameters"

_cancers, 2023, doi:10.3390/cancers15225372_

Round 1
Reviewer 1 Report
Comments and Suggestions for Authors
The manuscript by Nguyen et al. presented the recent updates on cancer diagnosis, prognosis, and monitoring via combining circulating tumor cells (CTCs) number and clinical parameters. Overall, the manuscript is interesting, and requires revision as follows:
Comments:
1. Lines 44-51, the information can be updated on cancer with minor details on molecular manifestation and pathways regulation, and recent various therapeutic strategies, i.e., https://doi.org/10.3390/biomedicines11061611.
2. Lines 110-118, please clearly state the objective and significance of this study with minor methodology.
3. Table 1, column 4 – please make a consistent presentation, as “space should be in between values and units”. Also, delete the term blood as it is common for all. These types of corrections should be followed at other places in the text or tables.
4. Add a new section on the challenges and perspectives.
5. Figures quality should be improved, i.e., font size and resolution, etc.
Comments on the Quality of English LanguageMinor editing of English language is required.
Author Response
We are grateful to the reviewers for their insightful comments on our review paper. We have revised the manuscript to reflect most of the suggestions provided by the reviewers. We have highlighted the changes within the manuscript.
Here is a point-by-point response to the reviewers’ comments and concerns.
Specific comment 1:
Lines 44-51, the information can be updated on cancer with minor details on molecular manifestation and pathways regulation, and recent various therapeutic strategies, i.e., https://doi.org/10.3390/biomedicines11061611.
Response 1:
Thank the reviewer for your suggestion indeed. We have added more information as suggested (Please refer to the manuscript of highlighted revision: page 2, lines 47-51 or below).
The occurrence of cancer involves the imbalance of many complex molecular mechanisms and regulatory pathways. Based on these pathogenic mechanisms, the progress of biomarkers and target drugs in recent years has indeed brought progress in the diagnosis and treatment of cancer.
Specific comment 2:
Lines 110-118, please clearly state the objective and significance of this study with minor methodology.
Response 2:
Thank you for pointing this out. We have revised this part to reflect your suggestion. (Please refer to the manuscript of highlighted revision: page 3, lines 114-121 or below).
In this review, the background information relevant to CTCs, including the characteristics of CTCs, the clinical use of CTC counting, and the technologies for CTC enrichment, were first introduced. This was followed by the discussion of challenges and new perspectives on applying the current CTC counting techniques for clinical practice. More importantly, this review discussed combining CTCs counts with the routinely used clinical data to provide patients with more appropriate and accurate diagnostic strategies when the technical challenges related to CTC research have not yet been solved.
Specific comment 3:
Table 1, column 4 – please make a consistent presentation, as “space should be in between values and units”. Also, delete the term blood as it is common for all. These types of corrections should be followed at other places in the text or tables.
Response 3:
Thank you for pointing this out. We have revised Table 1, column 4 according to your comment. (Please refer to the manuscript of highlighted revision: Table 1, page 7-8).
Specific comment 4:
Add a new section on the challenges and perspectives.
Response 4:
Thank you for this suggestion. It would have been interesting to explore this aspect. Therefore, we have added this section to the manuscript. (Please refer to the manuscript of highlighted revision: page 27, lines 421-442 or below).
In this review, we have discussed the technical advantages of combining the CTC counts with other conventional cancer-related parameters for analysis to overcome the limitation of the current CTC counting method. These combinations have been successfully demonstrated to improve the accuracy in early detection and prognosis evaluation of cancers. To obtain these clinical data for analysis, however, it could, to some extent, increase the economic burden for patients and medical institutions. In addition, an inappropriate combination of the clinical data might not be able to enhance the accuracy of cancer-relevant evaluation. Finding the appropriate combinations of clinical data, therefore, would be an important issue. This issue could be explored by effective data mining of big data, or with the aid of artificial intelligence (AI) and machine learning. Apart from the combination of other conventional cancer-related parameters for analysis, the data relevant to the CTC subtypes could provide clinically valuable information. To obtain these information, more sophisticated analytic equipment like imaging flow cytometry is required. Moreover, with the application of aptamers, research on designing different aptamers for the purpose of more specific and comprehensive CTC detection is also increasing. The application of DNA or RNA aptamers has already produced impressive results in lung and breast cancer (DOI: 10.1038/mt.2015.108 and DOI: 10.3389/fmolb.2023.1184285). Especially in the detection of CTCs in non-epithelial cell cancers such as glial brain tumors, the ap-plication of aptamers also has considerable potential (DOI: 10.1016/j.omtn.2023.03.015). Therefore, even though many challenges exist, the com-prehensive application of CTCs in cancer clinics is still expected to be achieved.
Specific comment 5:
Figures quality should be improved, i.e., font size and resolution, etc.
Response 5:
Thank you for this suggestion. We have modified the font size of words in the figure to improve the figure's quality. (Please refer to the figures in the manuscript: page 4, figure 1; page 20, figure 2 or below)

Reviewer 2 Report
Comments and Suggestions for Authors
It is well-organized and comprehensive review to include the information about their characteristics, clinical use of CTC counting, and technologies for CTC enrichment, followed by discussing the challenges and new perspectives of CTC counting techniques for clinical applications, and the advantages and the recent progress in combining CTC counts with other clinical parameter. Some points should be also noted as below.
1) We should be better add some new updates about cancer and CTC. A paper recently proposes that cancer is an is a complex ecological disease: a multidimensional spatiotemporal "unity of ecology and evolution" pathological ecosystem, and CTCs is also involved in such an ecological process (https://pubmed.ncbi.nlm.nih.gov/37056571/). This paper is suggested to be reviewed and make some discussion.
2) How about the different CTCs characteristics in early stage of and late stage of patients , e.g. molecular features, functional role of cancer colonization, the relation of these stages of CTCs and early/late DTCs.
3) Talk a little about CTCs immune and metabolism?
4) As to the current CTC counting techniques for clinical applications, which one do you think is the most ideal?
Author Response
We are grateful to the reviewers for their insightful comments on our review paper. We have revised the manuscript to reflect most of the suggestions provided by the reviewers. We have highlighted the changes within the manuscript.
Here is a point-by-point response to the reviewers’ comments and concerns.
Specific comment 1:
We should be better add some new updates about cancer and CTC. A paper recently proposes that cancer is an is a complex ecological disease: a multidimensional spatiotemporal "unity of ecology and evolution" pathological ecosystem, and CTCs is also involved in such an ecological process (https://pubmed.ncbi.nlm.nih.gov/37056571/). This paper is suggested to be reviewed and make some discussion.
Response 1:
Thank the reviewer for your suggestion indeed. We have added more information as suggested (please refer to the manuscript of highlighted revision: page 5, lines 177-181 or below).
Furthermore, the dynamic interplay between CTCs and their microenvironments is crucial. A tumor can be viewed as an integrated ecosystem where the co-evolution of neoplastic cells within the tumor microenvironment results in a wide range of cancer cell phenotypes. This is closely related to the tumor heterogeneity of CTCs.
Specific comment 2:
How about the different CTCs characteristics in early stage of and late stage of patients, e.g. molecular features, functional role of cancer colonization, the relation of these stages of CTCs and early/late DTCs.
Response 2:
Thank you for pointing this out. It would have been interesting to explore this aspect. We have added more information to emphasize this point (please refer to the manuscript of highlighted revision: page 5, lines 193-196 or below).
CTCs in early-stage cancer patients may retain the characteristics of primary tumors and tend to be more epithelial. However, in advanced-stage cancer patients, due to the cell metastasis mechanism of cells undergoing the EMT process, the pattern of CTCs becomes more mesenchymal type.
Specific comment 3:
Talk a little about CTCs immune and metabolism?
Response 3:
Thank you for this suggestion. It would have been interesting to explore this aspect. However, in the case of our review, CTC metabolism seems slightly out of scope because we focus on the application and challenge of CTC counting in cancer clinical practice. However, we have mentioned the CTC immunity issue when discussing the CTCs cluster in the manuscript (please refer to the manuscript of highlighted revision: page 16, lines 192-194 or below).
Among them, CTCs that heterogeneously combine with neutrophils to form clusters through neutrophil extracellular traps can escape immune surveillance by blocking peripheral leukocytes activation.
Specific comment 4:
As to the current CTC counting techniques for clinical applications, which one do you think is the most ideal?
Response 4:
You have raised an important point here. The general method for CTC counting mainly consists of three steps: (1) CTC enrichment, (2) CTC identification, and (3) CTC quantitation. For the most ideal CTCs counting techniques, in the manuscript, we have discussed the advantages and disadvantages of the current techniques for CTC enrichment (i.e., the first step). The current CTC enrichment techniques can be generally classified into two mechanisms: (i) based on the biophysical properties of CTCs (e.g., cell size, density, and bioelectrical properties) and (ii) based on the immune properties of CTCs (e.g., positive or negative immunoselection). Overall, we have mentioned that negative immunoselection-based can significantly improve the efficiency of CTC counting due to the potential recovery of all types of CTCs. All these discussions can be found in the manuscript (please refer to the manuscript of highlighted revision: page 15, lines 114-116).

Reviewer 3 Report
Comments and Suggestions for Authors
ChatGPT | FREE BOT, [28.10.2023 12:29]
⚠ Ошибка
Максимальный запрос 2000 символов
ChatGPT | FREE BOT, [28.10.2023 12:29]
This review analyzes the current status and clinical utility of CTC-based cancer diagnostic methods. The article is well-written and will be interesting for readers of the "Cancers" journal. Clinical applications are well-discussed, although in my opinion, it would be useful to include a short paragraph on specific protein biomarkers of CTCs (not only EpCAM) and provide more information on their biological role.
Some major comments:
1. CTCs are heterogeneous; therefore, to identify all CTCs, a complex of antibodies is required to cover the maximum number of antigens. However, the author only mentioned standard antibodies, such as EpCAM+, cytokeratin+, CD45-, and DAPI, as antibodies capable of detecting CTCs. What other antibodies can be used to search for and identify CTCs?
2. How can nonepithelial CTCs, particularly CTCs in malignant brain tumors, be identified, which is necessary for screening, early diagnosis, and monitoring of antitumor therapy?
3. The authors state that CTCs lack unique markers; however, there is evidence that the unique features of CTCs can be used to identify the primary tumor, as shown in studies: DOI: 10.1038/mt.2015.108 and DOI: 10.3389/fmolb.2023.1184285).
4. Aptamers, which are functional analogues of antibodies, have been successfully used to isolate and identify CTCs. However, the authors did not provide any examples of the use of aptamers for the determination of CTCs.
5. The methods for detecting CTC clusters are not clearly explained. How do these methods differ from the methods used to count single CTCs?
Minor:
Line 16, 27: The sentences in the Summary and Abstract are identical.
Table 1: Abbreviations such as AUC, OS, NE, MACS, and others are not explained.
Line 137: The use of cancer cell-specific DNA aptamers instead of well-known antibodies can improve some disadvantages associated with the limited number of CTC markers (see DOI: 10.1038/mt.2015.108,DOI: 10.1016/j.omtn.2023.03.015 and DOI: 10.3389/fmolb.2023.1184285).
Line 272: Some text has shifted from Figure 2 capture.
Lines 406-407: The sentence is duplicated.
Comments on the Quality of English LanguageThe English in the provided text appears to be of good quality. The sentences are grammatically correct, and the ideas are expressed clearly. There are no noticeable spelling or punctuation errors. Despite the correct grammar, sometimes language constructions are difficult to understand. Please note, I'm not a native speaker to judge the quality of English in details.
Author Response
We are grateful to the reviewers for their insightful comments on our review paper. We have revised the manuscript to reflect most of the suggestions provided by the reviewers. We have highlighted the changes within the manuscript.
Here is a point-by-point response to the reviewers’ comments and concerns.
Specific comment 1:
CTCs are heterogeneous; therefore, to identify all CTCs, a complex of antibodies is required to cover the maximum number of antigens. However, the author only mentioned standard antibodies, such as EpCAM+, cytokeratin+, CD45-, and DAPI, as antibodies capable of detecting CTCs. What other antibodies can be used to search for and identify CTCs?
Response 1:
Thank you for pointing this out. We have added more information to emphasize this point (please refer to the manuscript of highlighted revision: page 16, lines 184-187 or below).
In order to distinguish these different subtypes of CTCs, in addition to the traditional EpCAM antibodies, different antibodies such as vimentin or CD44 can also be used to classify CTCs with specific targets.
Specific comment 2:
How can nonepithelial CTCs, particularly CTCs in malignant brain tumors, be identified, which is necessary for screening, early diagnosis, and monitoring of antitumor therapy?
Response 2:
Thank you for pointing this out. We have added more information to emphasize this point (please refer to the manuscript of highlighted revision: page 27, line 439-440 or below).
Especially in the detection of CTCs in non-epithelial cell cancers such as glial brain tumors, the application of aptamers also has considerable potential.
Specific comment 3:
The authors state that CTCs lack unique markers; however, there is evidence that the unique features of CTCs can be used to identify the primary tumor, as shown in studies: DOI: 10.1038/mt.2015.108 and DOI: 10.3389/fmolb.2023.1184285).
Response 3:
Thank you for pointing this out. We have incorporated your suggestion and cited references in the manuscript (please refer to the manuscript of highlighted revision: page 27, lines 435-438 or below).
Moreover, with the application of aptamers, research on designing different aptamers for the purpose of more specific and comprehensive CTC detection is also increasing. The application of DNA or RNA aptamers has already produced impressive results in lung and breast cancer.
Specific comment 4:
Aptamers, which are functional analogues of antibodies, have been successfully used to isolate and identify CTCs. However, the authors did not provide any examples of the use of aptamers for the determination of CTCs
Response 4:
Thank you for pointing this out. We have incorporated your suggestion in the manuscript (please refer to the manuscript of highlighted revision: page 27, lines 439-440 or below).
Especially in the detection of CTCs in non-epithelial cell cancers such as glial brain tumors, the ap-plication of aptamers also has considerable potential.
Specific comment 5:
The methods for detecting CTC clusters are not clearly explained. How do these methods differ from the methods used to count single CTCs?
Response 5:
Thank you for pointing this out. We agree with this comment. Therefore, we have added more information to emphasize this point (please refer to the manuscript of highlighted revision: page 16, lines 196-200 or below).
Regardless of the type of CTC cluster, the detection mainly relies on microscopy-related applications. Compared with the detection of single CTCs, it requires more antibodies for operation and reduces the physical damage caused during detection, making it more difficult to detect CTC clusters.

Round 2
Reviewer 2 Report
Comments and Suggestions for Authors
The authors have fully answered my questions.
Reviewer 3 Report
Comments and Suggestions for Authors
The authors imploved the manuscript, and now it is ready for publishing.